# Torque Teno Virus DNA Load in Blood as an Immune Status Biomarker in Adult Hematological Patients: The State of the Art and Future Prospects

**DOI:** 10.3390/v16030459

**Published:** 2024-03-17

**Authors:** Eliseo Albert, Estela Giménez, Rafael Hernani, José Luis Piñana, Carlos Solano, David Navarro

**Affiliations:** 1Microbiology Service, Clinic University Hospital, INCLIVA Health Research Institute, 46010 Valencia, Spain; eliseo.al.vi@gmail.com (E.A.); estela.gimenez.quiles@gmail.com (E.G.); 2Centro de Investigación Biomédica en Red de Enfermedades Infecciosas, 28029 Madrid, Spain; 3Hematology Service, Hospital Clínico Universitario, INCLIVA Health Research Institute, 46010 Valencia, Spain; hernani_raf@gva.es (R.H.); jlpinana@gmail.com (J.L.P.); carlos.solano@uv.es (C.S.); 4Department of Medicine, School of Medicine, University of Valencia, 46010 Valencia, Spain; 5Department of Microbiology, School of Medicine, University of Valencia, 46010 Valencia, Spain

**Keywords:** anelloviruses, Torque teno virus (TTV), TTV DNA load in blood, allogeneic hematopoietic stem cell transplant recipients, hematological patients, opportunistic viral infection, graft versus host disease

## Abstract

A solid body of scientific evidence supports the assumption that Torque teno virus (TTV) DNA load in the blood compartment may behave as a biomarker of immunosuppression in solid organ transplant recipients; in this clinical setting, high or increasing TTV DNA levels precede the occurrence of infectious complications, whereas the opposite anticipates the development of acute rejection. The potential clinical value of the TTV DNA load in blood to infer the risk of opportunistic viral infection or immune-related (i.e., graft vs. host disease) clinical events in the hematological patient, if any, remains to be determined. In fact, contradictory data have been published on this matter in the allo-SCT setting. Studies addressing this topic, which we review and discuss herein, are highly heterogeneous as regards design, patient characteristics, time points selected for TTV DNA load monitoring, and PCR assays used for TTV DNA quantification. Moreover, clinical outcomes are often poorly defined. Prospective, ideally multicenter, and sufficiently powered studies with well-defined clinical outcomes are warranted to elucidate whether TTV DNA load monitoring in blood may be of any clinical value in the management of hematological patients.

## 1. Introduction

Anelloviruses (AVs) are nonenveloped icosahedral spherical viruses containing a circular, negative-sense, single-stranded DNA genome (ssDNA), ranging in size from 1.8 to 3.9 kb [1]. The AV genome includes one large open reading frame (ORF), named ORF1 or VP1, at least three more putative overlapping ORFs, (ORF2, ORF1/1, and ORF2/2), and an untranslated region (UTR), the latter including transcriptional regulatory elements, well conserved across species [1]. According to the recent taxonomic update by the International Committee on Taxonomy of Viruses (ICTV) [2,3], the family *Anelloviridae* currently comprises 30 genera and 155 species, this being indicative of its high genetic diversity, which is thought to be mainly driven by extensive recombination [4]. Of these genera, three commonly infect humans, *Alphatorquevirus*, *Betatorquevirus*, and *Gammatorquevirus*, and are conventionally referred to as Torque teno virus (TTV), Torque teno mini virus (TTMV), and Torque teno midi virus (TTMDV), respectively. The *Alphatorquevirus* genus includes 29 species (TTV 1 to 29), the *Betatorquevirus* genus comprises 12 species (TTMV 1 to 12), and the *Gammatorquevirus* genus includes two species (TTMDV 1 and 2) [2,3]. TTV is the most well-characterized AV and will thus be the focus of this review. The 29 TTV species are classified into five genogroups (50% nucleotide sequence divergence), which are divided into multiple genotypes with differences of up to 30% in the genome sequence. Virtually all of the human population is thought to be chronically infected with one or more AVs, which are the major eukaryotic virus constituents of the human virome, the so-called “anellome”. In fact, AVs are systematically detectable in the blood throughout life [5,6], as well as in many other body fluids (i.e., saliva, urine, feces, and bile) suggesting broad tropism [5] and, importantly, blood anellomes appear to be personal and compositionally stable over time [7]. Many aspects of AV biology and pathogenicity in humans remain to be elucidated. This is in part due to the lack of appropriate cell culture and animal models allowing efficient virus replication; currently, conclusive proof of AV involvement in human disease is missing.

Two previously published findings are relevant to the present review: (i) TTV appears to mainly replicate or persist in peripheral blood mononuclear cells and lymphocytes [8,9,10], and the detection of TTV DNA in monocytes and natural killer (NK) cells, as well as in B and T lymphocytes seems to support this assumption [8,9,10,11,12,13]; (ii) T-cell immunocompetence appears critical in the maintenance of a relatively stable anellome (discussed in [14]).

In a seminal study [15], De Vlaminck and colleagues suggested the potential application of the blood virome composition to predict the level of immunocompetence. A robust body of scientific evidence has been gathered over recent years, supporting the idea that high or increasing TTV DNA levels precede the occurrence of infectious complications in the solid organ transplantation (SOT) setting, whereas the opposite holds with the development of acute rejection. This supports the use of TTV DNA load in the systemic compartment as a reliable biomarker to assess the individual risk of infection and immune-related complications after SOT (reviewed in [16,17]). Unfortunately, the potential clinical value of the TTV DNA load in blood to gauge the risk of opportunistic infection or immune-related (i.e., graft vs. host disease; GvHD) clinical events in the hematological patient has not been firmly established. Here, we review and discuss the available scientific literature on this topic, especially focused on allogeneic hematopoietic stem cell transplant (allo-SCT) recipients.

## 2. Methodological Considerations

Most studies monitoring TTV (AV) DNA load kinetics in blood from hematological patients used PCR assays targeting different regions of the TTV genome. Early nested and heminested protocols amplified the N22 sequence of ORF1 [1]; these have been replaced by quantitative real-time PCR assays targeting the 5′-untranslated region (UTR). These are potentially capable of detecting the majority, if not all TTV species due to the high level of sequence conservation across TTV genogroups [18,19]. In 2017, a UTR-based (128 bp amplicon) PCR assay was marketed (TTV R-GENE kit; BioMerieux, Marcy-l’Étoile, France) for the detection and quantification of TTV DNA in plasma and whole blood samples by using 5′ nuclease TaqMan technology; the kit includes a range of four quantification standards. In a recent study [20], the analytical performance of the TTV-R GENE assay was compared with that of a laboratory-developed UTR-based real-time PCR assay and an in-house digital droplet PCR (ddPCR)**.** The three assays had comparably lower limits of TTV DNA detection (10–12 copies/mL) in blood samples, an overall agreement > 90% (most samples yielded TTV DNA loads within 0.5 log_10_ copies/mL), and were capable of similarly quantifying TTVs included in genogroups 1, 3, 4, and 5. In another recent study [21], degenerated primers targeting a highly conserved genomic sequence across all AVs were designed for genomic amplification and high-throughput sequencing of TTV, TTMV, and TTMDV, and a quantitative real-time PCR targeting conserved sequences across the three genera was developed. The study showed that codetection of TTV, TTMV, and TTMDV in both pretransplant and postengraftment plasma specimens was frequently documented (more than two-thirds of patients). Currently, metagenomic next-generation sequencing (mNGS) is being increasingly used as it allows a comprehensive exploration of the entire blood virome in an unbiased fashion [22]. In this context, Leijonhufvud et al. [23] analyzed a total of 22 blood samples from children with leukemia using real-time PCR and mNGS in parallel. They found that whereas 9/22 (41%) were positive by PCR, all had TTV reads with mNGS, thus suggesting that PCR assays with primers and probes designed to amplify conserved regions of the TTV genome may fail to detect some TTV strains. In fact, the authors found significant variant-associated differences in the sequences of the UTR primer site, affecting both primer efficiency and probe binding. mNGS is also being used to characterize the virome composition in other body fluids (e.g., bronchoalveolar specimens and cerebrospinal fluid samples; CSF) and its relationship with diseases in hematological and nonhematological patients [24,25]. In this context, Pou et al. [24] used mNGS to investigate the virome in CSF samples from 35 allo-SCT patients with neurological complications. They detected a significantly high number of reads mapping to TTV11, TTV13, and TTMDV in these patients compared with controls.

## 3. TTV DNA Load in Blood as a Biomarker of the Net State of Immunosuppression in Allo-SCT

### 3.1. Frequency of TTV DNA Detection in Allo-SCT Recipients

TTV DNAemia is commonly detected by laboratory-developed real-time PCR assays in patients with hematological disease before allo-SCT, although the frequencies reported vary widely between studies; this is likely related to differences in the limit of detection of the PCR assays employed. Masouridi-Levrat et al. [26], by using a real-time PCR assay, could detect TTV DNA in plasma from virtually all patients (121) at the time of transplantation. Albert et al. [27], also using a real-time PCR assay, found that 44 out of the 55 patients (80.0%) had detectable TTV DNAemia both before conditioning and at the time of allo-SCT, of whom 32 displayed TTV DNA loads within a range of 1.40–7.97 log_10_ copies/mL. Wohlfarth et al. [28] detected TTV DNA in plasma from 44/50 (80%) allo-SCT patients by real-time PCR before initiating conditioning therapy at a median level of 5.37 log_10_ copies/mL (IQR, 3.51–6.44 log_10_ copies/mL). Schmitz et al. [29] reported measurable TTV DNA by real-time PCR in whole blood from 62 out of 123 patients before initiating conditioning therapy. Spiertz et al. [30] detected TTV DNA in plasma by real-time PCR from 42 out of 54 (77.7%) patients at initial testing (Day −7 to +10) (median 3.14 log_10_ copies/mL); in turn, Zanella et al. [31] reported a lower rate of TTV DNA detection by real-time PCR (48%) at the time of allo-SCT. Discrepant results have been reported as to the impact of the underlying hematological disease and the modality of conditioning regimens on the magnitude of TTV DNA load before or at the time of transplantation. Masouridi-Levrat et al. [26] found higher TTV DNA loads among patients with acute lymphocytic leukemia and non-Hodgkin’s lymphoma, irrespective of the conditioning regimen, suggesting that lymphoid malignancies were associated with higher TTV DNA burden before allo-SCT, compared with those with myeloid malignancies. Albert et al. [27] reported that plasma TTV DNA loads at the time of allo-SCT were comparable, irrespective of the underlying hematological disease and type of conditioning. Wohlfarth et al. [28] also found that patients with lymphoid malignancies had significantly higher TTV DNA levels compared with those with myeloid malignancies; nevertheless, the conditioning regimen had no apparent effect on TTV DNA loads. Schmitz et al. [29] reported that patients with lymphoid malignancies had significantly more often detectable TTV levels (76.9%) than patients with myeloid diseases (43.3%; *p* < 0.05). Patients undergoing myeloablative conditioning had significantly higher TTV DNA loads than patients treated with reduced-intensity conditioning protocols (AUC days 50–100 and 0–300). The area under the curve (AUC) is calculated by the trapezoidal rule and is used as a cumulative measure of DNA loads over time.

Moreover, the use of high ATG doses in the conditioning regimen was associated with higher baseline TTV DNA loads. Notwithstanding, regardless of the underlying disease and conditioning regimen used, allo-SCT recipients display higher TTV DNA loads at the time of cell infusion than seemingly healthy control individuals [26,32,33]. Importantly, by day +100, virtually all allo-SCT recipients tested positive for TTV DNA in their blood. Table 1 summarizes the main findings of the aforementioned studies.

### 3.2. TTV DNAemia Kinetics in Allo-SCT Recipients

The following facts regarding the reconstitution of immunity after allo-SCT are pertinent to this subheading [39,40]. The cellular components of innate immunity, including monocytes, granulocytes, and NK cells, recover within weeks after allo-SCT, whereas the reconstitution of adaptive immunity proceeds at a slower pace. B- and T-cell counts normalize during the first months after transplantation, however, T-cell immunity, in particular, may remain functionally impaired for years. The speed at which immune reconstitution proceeds depends not only on the type of allograft but also several other factors, including the number of CD8^+^, CD3^+^, CD4^+^, NK cells in the cell infusion, the time from previous therapy to allo-SCT, or the type of conditioning regimen employed. Total CD3^+^ T-cell levels usually return to within the normal range by day +30. The recovery of CD8^+^ T-cells precedes that of CD4^+^ T-cells. CD4^+^CD45RO^+^ memory T-cell recovery occurs early after allo-SCT (2–3 weeks), whereas that of CD4^+^CD45RA^+^ naïve T-cells is delayed.

Data published on the kinetics of TTV DNAemia following allo-SCT tend to concur (depicted in Figure 1). TTV DNA load decreases after allo-SCT until engraftment, which appears to be linked to the depletion of lymphoid cells that occurs within the first weeks after transplantation [27,34]. Afterward, TTV DNA loads steadily rise reaching the peak 2–3 months after allo-SCT, this mirroring, to some extent, the reconstitution of the lymphoid cell pool [27,28,29,37]. Namely, Albert et al. [27] found the TTV DNA load to peak around day +90 (up to 7 log_10_ copies/mL), Wohlfarth et al. [28] at day 79 (up to 8 log_10_ copies/mL), Schmitz et al. [29] between days +90 and +105 (median, 5 log_10_ copies/mL), Spiertz et al. [30] at day +56 (median around 7 log_10_ copies/mL), and Pradier et al. [37] at day 100 (median 6.4 log_10_ copies/mL). Importantly, by day +100, virtually all allo-SCT recipients tested positive for TTV DNA. After day +100, the TTV DNA load slightly decreases, followed by a stable plateau over one year after transplantation [27,28,29,30,36].

The potential correlation between TTV DNA load in plasma and blood cell counts has been evaluated in several studies, yielding conflicting results. Albert et al. [27] reported that the magnitude of TTV DNA load between days +20 and +60 was directly correlated with absolute lymphocyte counts (ALCs), whereas it was inversely correlated between days +120 and +210 [27,36]; no correlation was reported between TTV DNA loads and white blood counts. The authors suggested that TTV DNA load may behave as a marker of T-cell reconstitution early after allo-SCT, while it may directly mirror the degree of immunosuppression after day +100. Supporting this assumption, the median TTV DNA AUC between days + 90 and +210 was significantly higher in patients under corticosteroid treatment for GvHD [36]. Nevertheless, Schmitz et al. [29] found no statistically significant correlation between TTV DNA load and peripheral counts of CD3^+^, CD3^+^/CD8^+^ suppressor T-cells, CD3^+^/CD4^+^ T-helper cells, or CD45^+^ lymphocytes within 365 days following transplantation. Pradier et al. [37] reported that at day +100, TTV DNA levels inversely correlated with the number of lymphocytes and, more specifically, with the number of CD4^+^ T and NK cells. Correlation between TTV DNA loads and CD4^+^ T-cells could be observed until day +300. No significant correlation with any cell subset was observed before day +100. Mouton et al. [32] found no significant correlation between TTV DNA load and ALC or CD3^+^ T-cell counts in 41 patients recruited around six months after allo-SCT. Differences in patient characteristics within the study cohorts and timing for TTV DNA load assessments may account for the discrepancies between studies. To interpret the above findings, it should be taken into consideration that the numeric repopulation of lymphocytes in allo-SCT does not necessarily reflect functional reconstitution. In this sense, Mouton et al. [32] found an inverse correlation between TTV DNA load and the proliferative ability of CD3^+^ T-cells but not with T-cell subtype counts, as stated above. In line with this observation, Focosi et al. [41] suggested that TTV DNA load is associated with the number of CD8^+^/CD57^+^ T-cells (markers of immunosenescence) in transplant recipients.

Interestingly, Albert et al. [35] detected TTV DNA more frequently in saliva than in plasma specimens before and at different time points after allo-SCT. Overall, TTV DNA loads were significantly higher in saliva than in plasma specimens (*p* = 0.0002) and correlated significantly (*p* ≤ 0.0001). The authors postulated that monitoring oral TTV DNA shedding may be useful for inferring immune reconstitution after allo-SCT.

### 3.3. TTV DNA Load and Occurrence of Clinically Relevant Events following Allo-SCT

There are conflicting observations regarding the relationship between the magnitude or dynamics of TTV DNA load in blood and the occurrence of clinically relevant events, including infection by opportunistic pathogens, immune-related events, such as GvHD, relapse of the underlying hematological disease, and overall survival (summarized in Table 2). Several factors may account for these discrepancies, including the baseline and post-transplant characteristics of the patients in the cohorts, timeframes at which such potential associations were explored, and, importantly, whether TTV DNA load was investigated as a predictive biomarker or assessed in patients that had already developed these clinical events.

#### 3.3.1. TTV DNA Load and Opportunistic Viral Infections

Some studies assessed the potential value of TTV DNA load as a biomarker anticipating the occurrence of certain viral infection events [30,33,38,42]. Albert et al. [42] investigated whether the early kinetics of plasma TTV DNAemia after allo-SCT were associated with the subsequent occurrence of Cytomegalovirus (CMV) and Epstein–Barr virus (EBV) DNAemia. The authors reported that the mean TTV DNA load AUC_20–30_ (between days +20 and +30 after allo-SCT) was lower in patients with subsequent CMV DNAemia (median, 3.3 log_10_ copies × days × mL^−1^) than in patients without (median, 4.4 log_10_ copies × day × mL^−1^), although the difference failed to reach statistical significance. Nevertheless, the mean TTV DNA load AUC_20–30_, however, not the mean ALC_20–30_, was significantly lower in patients with subsequent high-level CMV DNAemia requiring preemptive antiviral therapy following local guidelines (median, 2.7 log_10_ copies × days × mL^−1^) than in those without or who had no documented CMV DNAemia (median, 4.4 log_10_ copies × days × mL^−1^; range 0–8.43). A TTV DNA load AUC_20–30_ threshold level of 2.8 log_10_ copies × days × mL^−1^ had a predictive value of 64% for this event. Of relevance, a nonstatistically significant trend (*p* = 0.09) toward a direct correlation between TTV DNA AUC_20–30_ and peripheral counts of CMV pp65/IE-1-specific interferon-γ-producing CD8^+^ T-cells measured at day +30 after allo-SCT was seen. Of note, this dynamic parameter was not useful for anticipating the occurrence of EBV DNAemia. Supporting these findings, Spiertz et al. [30] observed that a TTV DNA load below 3 log_10_ copies/mL measured between Day −7 and +10 was significantly associated with a higher risk of developing CMV DNAemia. In turn, Forqué et al. [38] established TTV DNA load cut-offs (≥4.40 log_10_-pretransplant—and ≥4.58 log_10_-baseline—copies/mL) that predicted the occurrence of Polyomavirus BK hemorrhagic cystitis (BKPyV-HC) with a sensitivity of ≥89% and a negative predictive value of ≥96%. In turn, Gilles et. al. [33], in a small cohort (*n* = 23), showed that patients developing plasma CMV, EBV, or BKPyV DNAemia (one or more) within the first 100 days after allo-SCT displayed significantly higher (*p* = 0.005) TTV DNA loads (median, 9.26 log_10_ copies/mL) at day +30 than patients in whom these events were not documented (median, 6.40 log_10_ copies/mL). Nevertheless, virus DNAemia was often detected by real-time PCR before TTV DNA load monitoring (i.e., in 7 out of 12 patients with CMV DNAemia).

Other studies explored the potential association between TTV and opportunistic viral DNA loads but were not aimed at determining the potential anticipatory value of TTV DNA load measurements. For example, Wohlfarth et al. [28] reported a statistically significant, albeit weak to moderate, correlation between TTV and CMV DNA loads (Rho, 0.39, *p* < 0.01) and between TTV and EBV DNA loads (Rho, 0.45, *p* = 0.02) during phases of viral DNAemia. Nevertheless, while patients with and without CMV and EBV DNAemia had similar TTV loads throughout the study period, those with ongoing or preceding CMV DNAemia displayed higher TTV DNA levels at the end of follow-up. This contrasts somewhat with data published by Albert et al. [42], who found that the occurrence of CMV DNAemia did not have an apparent effect on TTV DNA load kinetics. Schmitz et al. [29] reported a similar AUC for log_10_ TTV-DNA loads quantified between days 0 and 50 and 0 and +300 after allo-SCT in patients with or without documented CMV and EBV DNAemia throughout the follow-up period. Pradier et al. [37] found that patients displaying higher TTV DNA levels at day +100 had higher rates of infection (undefined) at six months post transplant. Finally, Mouton et al. [32] reported higher TTV DNA loads in patients with CMV DNAemia than in those without (median, 4.8 vs. 3.7 log_10_ copies/mL; *p* = 0.02); in these series, allo-SCT patients were enrolled a median of six months post transplant, which likely implies that CMV DNAemia occurred before the TTV DNA load assessments.

#### 3.3.2. TTV DNA Load and Graft versus Host Disease

The potential use of TTV DNA load in blood as a predictive marker of acute (a) or chronic (c) GvHD has been investigated in several studies with conflicting results. Gilles et al. [33] showed that a TTV DNA load < 8.48 log_10_ copies/mL combined with a lymphocyte count ≥ 5.5 × 10^8^ cells/L at day +30 positively correlated with a low incidence of aGvHD within the first 100 days after allo-SCT. Wohlfarth et al. [28] found that TTV DNA load did not predict the occurrence of aGvHD or cGvHD; nevertheless, patients with aGVHD showed a trend toward higher TTV DNA loads measured on days +120 and +160. This difference reached statistical significance in patients who had not received ATG as part of the conditioning regimen. Schmitz et al. [29] compared TTV DNA loads between patients without aGvHD, with an aGvHD of grade II or higher, and with cGvHD, finding no significant differences. Pradier et al. [37] showed that patients with high TTV DNA loads at day +100 (>6.7 log_10_ copies/mL) tended toward a higher cumulative incidence of grades II-IV aGvHD at six months post transplant. Forqué et al. [38] found that TTV DNA loads > 3.38 log_10_ copies/mL at day +30 had a 90% sensitivity and 97% negative predictive value for anticipating aGvHD, whereas TTV DNA loads > 5.07 log_10_ predicted the occurrence of grades II-IV aGvHD with high sensitivity (83%), specificity (91%), and negative predictive value (98%).

The potential association between the presence or kinetics of AVs in stools and the occurrence of intestinal aGvHD (aIGvHD) has been assessed in two studies. Legoff et al. [43] longitudinally characterized the gut virome in 44 allo-SCT using mNGS. Increases in both the rates of detection (*p* < 0.0001) and number of sequences (*p* = 0.047) of AVs (among other persistent DNA viruses) over time were observed in individuals with enteric aGvHD relative to those without. Bueno et al. [44] hypothesized that quantification of TTV DNA load in stool specimens early after allo-HSCT may identify patients at high risk of aIGvHD. A total of 83 nonconsecutive adult allo-SCT patients were recruited, and the study period comprised the first 120 days after allo-HSCT. While median TTV DNA load values in post-transplant stool specimens were comparable (*p* = 0.34) in patients with or without subsequent aIGvHD, a decreasing trajectory (reduction in TTV DNA load > 0.5 log_10_ copies/0.1 g) in paired pretransplant and post-transplant specimens was independently associated with the occurrence of aIGvHD (OR, 5.2; 95% CI, 1.3–21.3; *p* = 0.02). Notably, a rising trajectory had a negative predictive value of 87.5% for aIGvHD.

#### 3.3.3. TTV DNA Load and Other Clinical Events

Pradier et al. [37] evaluated the TTV DNA load in a total of 58 allo-HSCT patients at day +100. They found a statistically significant association between higher TTV DNA loads (>6.70 log_10_ copies/mL) and worse two-year overall survival after adjusting by transplant and disease characteristics (adjusted HR, 3.51; *p* = 0.03). A trend towards an association between two-year progression-free survival (higher) and relapse rate (higher) was also observed in patients with >6.70 log_10_ copies/mL at day +100. In contrast, Schmitz et al. [29] did not find an association between TTV DNA loads or different TTV DNA AUCs and overall mortality and relapse rates within the first year after allo-SCT.

## 4. Potential Clinical Value of TTV DNA Load Assessment in Other Hematological Patients

Focosi et al. [45] investigated plasma TTV DNA load kinetics in hematological patients undergoing autologous hematopoietic stem cell transplantation. The authors found that, independent of the underlying hematological disease and therapeutic regimen, the TTV DNA load increased after transplantation in parallel with circulating CD8^+^ CD57^+^ T lymphocytes, known to represent an indirect marker of functional immune deficiency, and suggested that TTV DNAemia kinetics may serve as a surrogate marker of functional immune reconstitution in this clinical setting.

Chimeric antigen receptor T-cell (CAR-T) therapy has improved outcomes in relapsed/refractory (R/R) patients with hematological malignancies, including B-acute lymphoblastic leukemia, large B-cell lymphoma (LBCL), follicular lymphoma, mantle cell lymphoma, and multiple myeloma [46]. CARs are bioengineered hybrid receptors that exhibit antigen specificity for monoclonal antibodies. These receptors are artificially introduced and overexpressed in autologous T-cells, enabling the recognition of tumor surface antigens, thus bypassing the need for major histocompatibility complex-restricted antigen presentation [47]. In the week preceding CAR-T infusion, patients undergo lymphodepleting chemotherapy (LC), typically involving fludarabine and cyclophosphamide or bendamustine alone, to enhance CAR-T expansion [48]. However, both LC regimens are lymphotoxic, and together with the toxicity of prior treatments, this implies that immune competence is compromised even before the CAR-T infusion itself [49,50,51]. Wudhikarn et al. reported a median ALC of 600/mm^3^ (0–2500) before LC, with almost half of the patients (48.3%) displaying an ALC < 500/mm^3^ [49], consistent with more recent studies [50,51]. Following CAR-T infusion, immune reconstitution may be hampered not only by immune toxicities (such as cytokine release syndrome or neurotoxicity) but also by their treatment, commonly corticosteroids and/or tocilizumab [52]. Altogether, this may increase the risk of secondary infections, further impacting immune reconstitution [53].

After an early decrease, the ALC gradually rebounds, with 60% of patients achieving an ALC > 500/mm^3^ six months after CAR-T infusion [49]. Concerning CD4^+^ T-cells, significant recovery is not observed during the first year of therapy [49,50,54], with >50% of responders having less than 200/mm^3^. In terms of CD19^+^ cells, nearly 40% of patients already exhibit B-cell aplasia and hypogammaglobulinemia before CAR-T therapy [50,54,55]. Following CAR-T infusion, CD19^+^ cells and IgG levels decline, reaching a nadir at one and six months, respectively. Despite gradual recovery, 40% of responders maintain B-cell aplasia one year after CAR-T therapy [51,54]. In this context, Benzaquén et al. characterized the dynamics of TTV DNA load in 79 adult patients receiving CAR-T therapy for R/R LBCL [51]. The TTV DNA load decreased after LC until day 10, followed by a steady increase until reaching the maximum load around day +90. A TTV DNA load < 4.05 log_10_ copies/mL at neurotoxicity onset identified patients at risk of severe neurotoxicity (OR, 16.68, *p* = 0.048). Patients with falling or stable TTV DNA loads had better progression-free survival than those with ascending TTV DNA loads (HR, 0.31, *p =* 0.006). The authors suggested that TTV monitoring could serve as a surrogate marker of immune competence, predicting CAR-T efficacy and toxicity, potentially guiding therapeutic strategies.

Molecular targeting agents, including Bruton Tyrosine Kinase inhibitors (e.g., ibrutinib) and intracellular Janus kinase (JAK) signal transducer, and activators of transcription (STAT) pathway inhibitors (e.g., ruxolitinib) are first-line drugs used to treat several hematological malignancies [56]. These drugs have a profound effect on B- and T-cell homeostasis, which largely explains the increased incidence of opportunistic infections in patients treated with these compounds [57]. Solano de la Asunción et al. [58] characterized the kinetics of TTV DNA load in plasma (at baseline and days +15, +30, +45, +60, +75, +90, +120, +150, and +180 after treatment administration) in patients treated with ibrutinib or ruxolitinib, and investigated whether TTV DNAemia dynamics could anticipate CMV DNAemia, a common event in these patients [59,60]. TTV DNA load increased over time in ibrutinib-treated patients, reaching the peak by day +120. There was a moderate inverse correlation between TTV DNA load and ALC (however, not with the CD4^+^ or CD8^+^ T-cells analyzed separately). In ruxolitinib-treated patients, the authors observed a trend toward increasing TTV DNA loads, peaking by day +60 after treatment.

TTV DNA loads quantified at day +30 in both ibrutinib- and ruxolitinib-treated patients were not associated with the subsequent development of CMV DNAemia.

No correlation was observed between TTV DNA loads and CMV-specific IFN-γ-producing CD8^+^ and CD4^+^ T-cell counts in either patient group. Thus, the data refuted the hypothesis that TTV DNA load monitoring in hematological patients treated with either ibrutinib or ruxolitinib could be useful in predicting the occurrence of CMV DNAemia or the level of CMV-specific T-cell reconstitution.

## 5. Future Prospects

There is solid evidence supporting the use of TTV DNA load as an ancillary biomarker for predicting the occurrence of opportunistic viral infection and immune-related events, such as organ rejection in SOT, thus providing useful information for immunosuppression therapy guidance. However, this is not the case in allo-SCT recipients or nontransplant hematological patients, such as those undergoing CAR-T cell therapy or treated with small molecule inhibitors. In fact, contradictory data have been published as to the association between TTV DNA load kinetics in the blood compartment and the occurrence of viral infections or acute or chronic GvHD in allo-SCT recipients. Studies addressing this topic are highly heterogeneous in design, with only a few exploring the potential value of TTV DNA load for predicting clinical events. They are frequently of small size, with fairly dissimilar patient cohorts and TTV DNA load monitoring time points; they use different PCR assays for TTV DNA quantification and often have poorly defined clinical outcomes. To further complicate matters, TTV immunobiology in hematological patients (in particular, allo-SCT recipients) is apparently much more complex than that in SOT. In fact, due to the kinetics of immune cell repopulation after allo-SCT and the apparent dependence of TTV on lymphocytes for replication, TTV DNA may likely behave as a marker of immunocompetence early after engraftment and as a marker of immunosuppression later on (after day +100). Thus, prospective, ideally multicenter, and sufficiently powered studies with well-defined clinical outcomes are warranted to elucidate whether TTV DNA load monitoring in blood may be of any clinical value in the management of hematological patients.

## Figures and Tables

**Figure 1 viruses-16-00459-f001:**
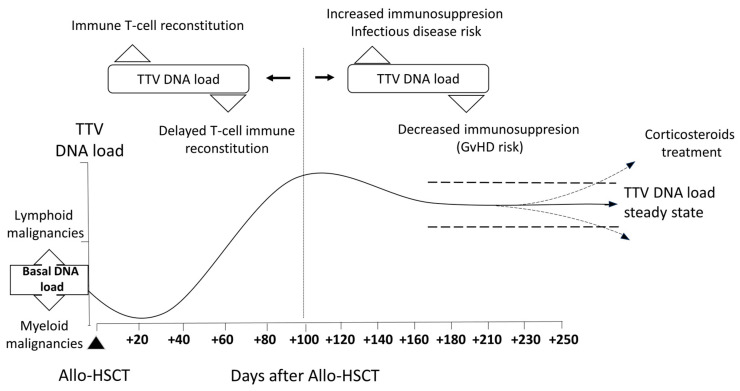
Proposal for TTV DNA load kinetics in allogeneic hematopoietic stem cell transplant recipients and clinical factors that may impact it. “Pretransplant TTV DNA loads may be affected by the underlying disease. After SCT, TTV DNA loads decrease in parallel with lymphodepletion, and then increase until day +100, reaching a peak and behaving as a marker of immune T-cell reconstitution. After day +100, TTV DNA loads inversely correlate with immune functionality, serving as a risk marker for GvHD and infectious events. Steady TTV DNA loads may be affected by corticosteroid treatment”.

**Table 1 viruses-16-00459-t001:** Representative studies assessing the kinetics of Torque teno virus DNAemia in blood from adult allogeneic hematological stem cell transplant recipients.

Study	Sample Size and Underlying Disease	Monitoring Time Points	Main Findings
Maggi et al. (2010) [34]	Four patients with acute leukemia	Before total body irradiation (TBI) and after allo-SCT (within the first 30 days; in 2 patients at days +50, +80, and +110 after transplantation)	All patients (*n* = 4) developed plasma TTV DNAemia before TBI and 3 after allo-SCT (day +30) TTV DNA load before TBI, median, 5.15 log_10_ copies/mL and median, 5.4 log_10_ copies/mL at day +80. Starting from day +26, TTV DNAemia consistently increased in parallel with white blood cells in peripheral blood.
Masouridi-Levrat et al. (2016) [26]	A total of 121 patients. AML (58), ALL (15), MDS (12), NHL (10), MPS (6), MM (9), HL (5), CML (3), CLL (1), and MDPS (1)	At baseline (days +4.3 ± 5.4 post-transplant) and at 1 and 3 months afterward.	At baseline, 85 out of 121 patients had detectable TTV DNAemia, and all patients (77 out of 77) who were still available for follow-up had detectable TTV DNA at the end of it. At baseline, TTV DNA load was 2.38 log_10_ copies/mL (median) and 5.47 log_10_ copies/mL (median) 3 months later. TTV DNA load after transplantation was higher in patients who received immune suppression, including prednisone. At baseline, TTV DNA load was higher in patients with ALL or NHL
Albert, et al. (2017) [27]	A total of 72 patients. HL (5), NHL (15), ALL (7), CLL (6), AML (19), CML (1), MM (5), MDS (10), and others (4)	Before conditioning, and at days +20, +30, +60, and +90 after allo-SCT.	Before conditioning 44/55 had detectable DNA load. A total of 23/23 patients with paired samples between days +20 and +90 had detectable TTV DNA in all available samples. TTV DNA load reached peak levels at around day +90 (median, 5.0 log_10_ copies/mL) TTV DNA load in plasma decreased dramatically after conditioning, in parallel with absolute lymphocyte counts. TTV DNA load increased steadily after engraftment, in parallel with absolute lymphocyte counts (rho = 0.21; *p* = 0.09)
Wohlfarth et al. (2018) [28]	A total of 50 patients. ALM (25); ALL (9) MDS (6), NHL (3), and others (7)	Before conditioning, at the time of allo-SCT and days +10, +30, +50, +80, +120, +160, +200,+ 250, +300, and +365 after allo-SCT	A total of 40/50 patients (80%) had detectable TTV DNA loads before conditioning and all developed plasma TTV DNAemia during the follow-up period TTV DNA load peaked (8.32 log_10_ copies/mL) at around day +80 TTV DNA levels showed a weak yet significant inverse correlation with absolute lymphocyte counts following engraftment (rho = −0.27; *p* < 0.01) TTV DNA load reached a stable plateau towards the end of the follow-up period.
Albert et al. (2018) [35]	A total of 38 patients. NHL (14), HL (1), AML (4), CLL (3), AML (6), MM (2), MDPS (6), and others (2)	Before the initiation of conditioning (day −7) and days +30, +50, +90 after allo-SSCT.	At baseline (pre-transplant), TTV DNA was detected in all saliva samples (23/23) and all but 3 plasma samples (20/23) At day +90, all saliva samples (25/25) and all but one plasma specimen (24/25) had detectable TTV DNA. The kinetics of TTV DNA load in saliva and plasma specimens were comparable. The TTV DNA load peak was reached at day +90 in both compartments (median, 6.1 log_10_ copies/mL in saliva and 4.7 log_10_ copies/mL in plasma). A direct correlation between absolute lymphocyte counts and TTV DNA loads in plasma was seen after engraftment. (rho = 0.507; *p* < 0.0001)
Albert et al. (2019) [36]	A total of 33 patients: Lymphoma (12), leukemia (12), MM (4), MDS (3), and myelofibrosis (2)	Before conditioning and at days +20, +30, +40, +50, +60, +90, +120, +180, +210 after allo-SCT	Detection rate over the study period 100% (33/33). Increasing TTV DNA levels after engraftment peaking at + 90 (median, 5.1 log_10_ copies/mL) TTV DNA loads measured within days +120 and +210 correlated inversely with absolute lymphocyte counts (rho = −0.26; *p =* 0.003)
Mouton et al. (2020) [32]	A total of 41 patients. Myeloid neoplasm and acute leukemia (37), Mature lymphoid/histiocytic, and dendritic neoplasms (4)	Patients were enrolled a median of 6 months (IQR, 5–8) post transplant.	DNA TTV was detected in 100% (41/41) of patients. Peak TTV DNA load was 3.9 log_10_ copies/mL. There was no significant correlation between TTV DNA load and absolute lymphocyte counts or CD3^+^ T-cells. TTV DNA load was inversely correlated with CD3^+^ T-cell proliferation capacity (%) (rho = −0.39, *p* = 0.01)
Schmitz et al. (2020) [29]	A total of 123 patients. AML (58), ALL (9), MDS (33), NHL (11), and others (12)	A total of 18 different time points were examined from before allo-HSCT to 345 days post.	A total of 62/123 (50.4%) had measurable TTV DNA before transplantation. Peak TTV DNA was reached at day +90 (median, 5 log_10_ copies/mL). No statistically significant correlation was found between TTV DNA load and any lymphocyte subset. Patients with lymphatic malignancies had significantly more often detectable TTV DNA before allo-SCT compared with patients with myeloid diseases.
Pradier et al. (2020) [37]	A total of 168 patients. AML (78), ALL (17), MDS (22), MPS (11), Lymphoma (12), MM (11), and others (17).	Peripheral blood samples were collected at days 0, +50, +100, +150, +200, +300, +400, +547, and 2 to 9 years post-allo-SCT.	TTV titer reached a peak value at day +100 (median 6.4 log_10_ copies/mL, IQR 5.1–7.7) At day 100, there was an inverse correlation between TTV levels and lymphocyte counts, particularly with CD4^+^ T-cells and NK cells. The correlation between TTV and CD4^+^ T-cell counts persisted until day 300. No significant correlation with any cell subset was observed before day 100 or after day 400.
Spiertz et al. (2023) [30]	A total of 59 patients. AML (31) ALL (6), CML (2), CLL (4), MDS (10), and others (6)	Upon infusion of hematopoietic cells (between day −7 and +10), and at days +14, +21, +28, +56, +90, +365 after allo-SCT	The prevalence of TTV DNA detection at baseline was 77% 42(54). All patients tested positive for TTV DNA on day +90. Furthermore, 95.6% of the samples remained TTV DNA positive one year after allo-SCT TTV DNA load peak was reached at day +56 (7.08 log_10_ copies/mL)
Forque et al. (2023) [38]	A total of 75 patients. AML (28), H (13) NHL (15) MDS (4) MF (4), CML (3), CLL (3), and others (5)	Before conditioning, at baseline, and after allo-SCT (+30, +60, +90, +120, and +180)	A total of 73/75 patients had detectable TTV DNA load (97%) over the study period. The TTV DNA load decreased by day +30 compared with that at baseline (a reduction of 15%), and then increased, reaching its peak by day +90 with a median load of 7.29 log_10_ copies/mL.

ALL, acute lymphocytic leukemia; Allo-SCT, allogeneic hematological stem cell transplant recipient; AML, acute myeloid leukemia; CLL, chronic lymphocytic leukemia; CML, chronic myelocytic leukemia; HL, Hodgkin lymphoma; MDS, myelodysplastic syndrome; MDPS, myelodysplastic/myeloproliferative syndrome; MM, multiple myeloma; MPS, myeloproliferative syndrome; NHL, non-Hodgkin lymphoma; TTV, Torque teno virus.

**Table 2 viruses-16-00459-t002:** Association between TTV DNA load in blood and the occurrence of viral opportunistic infections and graft versus host disease in allogeneic hematopoietic stem cell transplantation.

Study	Sample Size and Underlying Disease	Relevant Monitoring Time Points before or after Allo-SCT	Main Findings
Albert E et al. (2018) [42]	A total of 72 patients. HL (5), NHL (15), ALL (7), CLL (6), AML (19), CML (1), MM (5), MDS (10), and others (4).	Days +20, +30, +40, +50 after allo-SCT	Association between TTV DNA AUC_20–30_ and the risk of subsequent CMV or EBV DNAemia TTV DNA AUC_20–30_ ≤ 2.8 log_10_ copies × days × mL^−1^ associated with an increased risk of subsequent CsCMVi
Legoff et al. (2017) [43]	A total of 44 patients. Underlying disease not reported.	Longitudinal follow-up of the enteric virome (by metagenomics)	*Anelloviridae* reads were not predictive of enteric GvHD
Gilles et al. (2017) [33]	A total of 23 patients. AML (10), Lymphoma (7), MDS (1), Myelofibrosis (1), MM (1), CLL (1), prolymphocytic leukemia (1), chronic myelomonocytic leukemia (1) Low risk (9), High risk (14)	Days +30, +100, and +200	TTV DNA load on day +30 < 8.48 log_10_ copies/mL associated with lower incidence and severity of infectious complications (viral reactivations) Higher TTV DNA loads on day +30 associated with an increased risk of GvHD during the first 100 days after transplantation
Schmitz et al. (2020) [29]	A total of 123 patients. AML (58), ALL (9), MDS (33), NHL (11), and others (12)	Days + 0–15, +16–30, +31–45, +46–60, +61–80, +81–99, +100–119, +120–140, +141–160, +161–180, +181–200, +201–219, +221–239, +240–260, +261–280, +281–300, +301–320, +321–345	No differences in TTV DNA loads between patients with or without herpes virus reactivations No differences in TTV DNA load in patients with or without GvHD at the time of sampling
Wohlfarth et al. (2018) [28]	A total of 50 patients. ALM (25); ALL (9) MDS (6), NHL (3), and others (7)	Days +10, +30, +50, +80, +120, +160, +200,+ 250, +300 and +365	Higher TTV DNA load on days +120 and +160 in patients with GvHD who did not receive ATG in the conditioning. TTV DNA levels were not predictive of GvHD
Mouton W et al. (2020) [32]	A total of 41 patients. Myeloid neoplasm and acute leukemia (37), Mature lymphoid, histiocytic, and dendritic neoplasms (4)	Five to eight months post transplant	Higher plasma TTV DNA loads at 6 months after allo-SCT in patients with previous viral opportunistic infections/reactivations.
Pradier et al. (2020) [37]	A total of 133 patients. AML (78), ALL (17), MDS (22), MPS (11), Lymphoma (12), Myeloma (11), and others (17)	Days +50, +100, +150, +200, +300, +400, +547, and 2 to 9 years post-allo-SCT.	A total of >6.075 log_10_ copies/mL on day +100 associated with a higher risk of acute grade II-IV GvHD
Bueno et al. (2021) [44]	A total of 83 patients. ALL (49), CL (4), Lymphoma (4), MDS (19), Myelofibrosis (5), and MM (2)	TTV DNA was quantified in paired stool and plasma samples collected a median of 2 days before cell infusion and a median of 14 days after allo-HSCT by real-time PCR.	A Descendent trajectory between pre- and post-transplant stool samples was associated with the risk of subsequent intestinal GvHD
Spiertz et al. (2023) [30]	A total of 59 patients. AML (31), ALL (6), CML (2), CLL (4), MDS (10), and others (6)	Days +14, +21, +28, +56, +90 and +365	TTV DNA load determined upon allo-SCT (−7 to +10) below 3 log_10_ copies/mL was significantly associated with an increased incidence of CMV infection in multivariate analysis
Forque et al. (2023) [38]	A total of 75 patients. AML (28), HL (13), NHL (15), MDS (4), MF (4), CML (3), CLL (3), and others (5)	Preconditioning, at baseline (day 0) and days after allo-SCT (+30, +60, +90, +120, and +180)	Preconditioning and baseline TTV DNA loads (>4.40 and >4.58 log_10_ copies/mL respectively) predicted early (within the first 30 days) BKPyV-HC with high sensitivity (89–100%) and 65% specificity. TTV DNA load > 3.38 log_10_ copies/mL on day +30 associated with subsequent occurrence of all grades GvHD; TTV DNA load > 5.07 log_10_ copies/mL on day +30 associated with the development of II-IV GVHD

## Data Availability

Not applicable.

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
