# Peer review of "Torque Teno Virus DNA Load in Blood as an Immune Status Biomarker in Adult Hematological Patients: The State of the Art and Future Prospects"

_viruses, 2024, doi:10.3390/v16030459_

Round 1

Reviewer 1 Report

Comments and Suggestions for Authors

The authors have written a review article about the TTV genomic load as a bio marker for immune statues of haemato- oncological disease. The viral load could be suitable way that reflect on the immune status of those patient (immune suppression incidence) that could be indication for organ repulsion or incidence of other viral infection. the review was written in a pretty good manner, so it was difficult to catch some mistakes.

There is a variation in the used method for virus detection in the different samples. it was like narration for other people work. from this data authors could conclude the best method for TTV genomic detection. As it was noticed that the mNGS was the better test for determination of the viral load. so, if is possible for the authors to the test that they used for detection of TTV genomic load.    

In TTV DNA load and opportunistic viral infections paragraph in line 234 the authors use area under the curve (AUC) as expression for virus load could you use the log 10 to make it easy for understanding.

Could you please add more details in figure (1) legend? It will be easier for reader to understand easily. 

Author Response

The authors have written a review article about the TTV genomic load as a bio marker for immune statues of haemato- oncological disease. The viral load could be suitable way that reflect on the immune status of those patient (immune suppression incidence) that could be indication for organ repulsion or incidence of other viral infection. the review was written in a pretty good manner, so it was difficult to catch some mistakes.

We thank the reviewer’s comments.

There is a variation in the used method for virus detection in the different samples. it was like narration for other people work. from this data authors could conclude the best method for TTV genomic detection. As it was noticed that the mNGS was the better test for determination of the viral load. so, if is possible for the authors to the test that they used for detection of TTV genomic load.    

As we pointed out, the use of metagenomic next-generation sequencing (NGS) procedures enable a comprehensive assessment of the entire virome present in a given specimen. Metagenomic or targeted NGS allows the detection of beta and gamma anelloviruses in addition to Torque teno viruses, the latter usually targeted by qPCR. In the methodological considerations (page 2), all qPCRs (in house and commercially available) used for TTV quantification are reviewed.

In TTV DNA load and opportunistic viral infections paragraph in line 234 the authors use area under the curve (AUC) as expression for virus load could you use the log 10 to make it easy for understanding.

We have amended the text to show all the results in log 10 units. Also, a detailed explanation of AUC is provided in page 4: “The Area under the curve (AUC) is calculated by the trapezoidal rule and is used as a cumulative measure of DNA loads over time”.

Could you please add more details in figure (1) legend? It will be easier for reader to understand easily. 

We extended the legend: “Pre-transplant TTV DNA loads may be affected by the underlying disease. After SCT, TTV DNA loads decrease in parallel with lymphodepletion, and then increase until day +100 reaching a peak and behaving as a marker of immune T-cell reconstitution. After day +100, TTV DNA loads inversely correlate with immune functionality, serving as a risk marker for GvHD and infectious events. Steady TTV DNA loads may be affected by corticosteroid treatment”.

Reviewer 2 Report

Comments and Suggestions for Authors

The authors have written a review on the value of TTV DNA load determination in patients with hemato-oncologic disease. While quantitative TTV DNA determination in solid organ transplant patients reflects the degree of immunosuppression and is indicative of impending organ rejection or opportunistic infection, the authors point to contradictory data in patients after stem cell transplantation. They therefore recommend conducting multicenter prospective studies with appropriate case numbers to clarify the significance of TTV DNA load determination in hemato-oncological patients.

The review is very well written and of high clinical relevance. I only have a few aspects to comment on.

The abbreviation VP1 is not explained in the introduction. It probably refers to the capsid protein VP1 encoded on ORF1 (page 1).

Perhaps it would also be better to speak of PCR tests that detect different areas of the TTV genome(s) rather than PCR tests that detect different TTV sequences (page 2). The different tests also show that there is a need for standardization. At the very least, individual patients should always be monitored with the same test.

The specification of viral loads should be uniform. Sometimes this is given in copies/ml, then again in log10 copies/ml. I prefer a uniform specification in log10 copies/ml to ensure comparability.

Sometimes the viral load is also given in AUC log10 copies x days/ml. This information, which is still not widely used, is difficult for the reader to understand. I recommend a detailed explanation of this information, perhaps even with the help of an illustration.

Section 3.2 on the kinetics of TTV DNA after allogeneic stem cell transplantation lacks literature data on the reconstitution of the individual lymphocyte populations (page 8).

Figure 1 should be explained in detail in the legend so that it can be understood without reading the main text. For example, it is not clear to me from the figure alone why patients with lymphoid malagnancies have a higher TTV DNA load than patients with myeloid diseases (marked on the ordinate).

Author Response

The authors have written a review on the value of TTV DNA load determination in patients with hemato-oncologic disease. While quantitative TTV DNA determination in solid organ transplant patients reflects the degree of immunosuppression and is indicative of impending organ rejection or opportunistic infection, the authors point to contradictory data in patients after stem cell transplantation. They therefore recommend conducting multicenter prospective studies with appropriate case numbers to clarify the significance of TTV DNA load determination in hemato-oncological patients.

The review is very well written and of high clinical relevance. I only have a few aspects to comment on.

We thank the reviewer’s comments

The abbreviation VP1 is not explained in the introduction. It probably refers to the capsid protein VP1 encoded on ORF1 (page 1).

The large ORF that is common among anelloviruses is either called ORF1 or VP1.

Perhaps it would also be better to speak of PCR tests that detect different areas of the TTV genome(s) rather than PCR tests that detect different TTV sequences (page 2). The different tests also show that there is a need for standardization. At the very least, individual patients should always be monitored with the same test.

We have modified the sentence to “different areas of the TTV genome”, as suggested. We already pointed out that a PCR assay was marketed for quantification of TTV load, and that the analytical performance of the laboratory-developed assays was highly comparable.

The specification of viral loads should be uniform. Sometimes this is given in copies/ml, then again in log10 copies/ml. I prefer a uniform specification in log10 copies/ml to ensure comparability.

It has been changed accordingly.

Sometimes the viral load is also given in AUC log10 copies x days/ml. This information, which is still not widely used, is difficult for the reader to understand. I recommend a detailed explanation of this information, perhaps even with the help of an illustration.

We have added the following sentence for clarification: “The Area under the curve (AUC) is calculated by the trapezoidal rule and is used as a cumulative measure of DNA load over time”. We consider that a graphical explanation of AUC is not within the purposes of this review.

Section 3.2 on the kinetics of TTV DNA after allogeneic stem cell transplantation lacks literature data on the reconstitution of the individual lymphocyte populations (page 8).

References 34 and 35 discuss the kinetics of specific cellular subpopulations after allo-SCT.

Figure 1 should be explained in detail in the legend so that it can be understood without reading the main text. For example, it is not clear to me from the figure alone why patients with lymphoid malagnancies have a higher TTV DNA load than patients with myeloid diseases (marked on the ordinate).

Pre-transplant TTV DNA load can be influenced by the underlying disease. Some studies have shown higher TTV DNA loads in lymphoid than in myeloid malignancies. We have extended the Figure caption: “Pre-transplant TTV DNA loads may be affected by the underlying disease. After SCT, TTV DNA loads decrease in parallel with lymphodepletion, and then increase until day +100 reaching a peak and behaving as a marker of immune T-cell reconstitution. After day +100, TTV DNA loads inversely correlate with T-cell immune system, serving as a risk marker for GvHD and infectious events. Steady TTV DNA loads may be affected by corticosteroid treatment”.

Reviewer 3 Report

Comments and Suggestions for Authors

The manuscript by Albert et al provides an overview of the studies performed via TTV quantification in hematological patients, treated either by allogeneic hematological stem cell transplantation (allo-SCT) or chimeric antigen receptor T-cell (CAR-T)  therapy. The manuscript provides a nice overview, with the various studies clearly presented. The main outcome is that hematological patients treated by all-SCT or CAR-T show a less clear TTV-effect, which is different from solid organ transplant recipients - where TTV DNA quantification can be used as a predictive marker to guide immunosuppressive treatment. The complicated outcome in allo-SCT and CAR-T is most likely caused by the cell tropism of the virus, which could very well be T-cells, and as these cells are also part of the therapy, it complicates TTV-measure-predictions. It is very nice that the complexity is presented in this overview. Below a few suggestions for improvement:

-          Introduction section page 2, lines 58-66. In this paragraph four “facts” are presented, which are mentioned as “relevant for the present review”. It is strange to present these findings as “facts”, as for none of these four there is solid proof (e.g. via a robust culturing system), suggestion is to present these points as “previous findings”.

-          The second point ((“(ii)” in the paper, line 61) concerns the amount of TTV virions generated per day being 3.8 x 1010 per day. This number of produced virions per day has been calculated in one study on HCV infected patients only, and may require some extra study (e.g. in healthy persons). Suggestion is to delete this sentence as the amount of virus particles produced daily is not of relevance for the review.

-          The fourth point (“(iv)”, line 64) concerns compartmentalization. The studies referred to are quite old, and conducted at a time when the full width of anellomes were not known yet. It is still greatly unknown (as it is not carefully looked at) if compartmentalization truly occurs for human anelloviruses. Suggestion is to delete this sentence on compartmentalization as this is not of relevance for the review.

Author Response

The manuscript by Albert et al provides an overview of the studies performed via TTV quantification in hematological patients, treated either by allogeneic hematological stem cell transplantation (allo-SCT) or chimeric antigen receptor T-cell (CAR-T)  therapy. The manuscript provides a nice overview, with the various studies clearly presented. The main outcome is that hematological patients treated by all-SCT or CAR-T show a less clear TTV-effect, which is different from solid organ transplant recipients - where TTV DNA quantification can be used as a predictive marker to guide immunosuppressive treatment. The complicated outcome in allo-SCT and CAR-T is most likely caused by the cell tropism of the virus, which could very well be T-cells, and as these cells are also part of the therapy, it complicates TTV-measure-predictions. It is very nice that the complexity is presented in this overview. Below a few suggestions for improvement:

We thank the reviewer’s comments

-      Introduction section page 2, lines 58-66. In this paragraph four “facts” are presented, which are mentioned as “relevant for the present review”. It is strange to present these findings as “facts”, as for none of these four there is solid proof (e.g. via a robust culturing system), suggestion is to present these points as “previous findings”.

We have followed the reviewer’s suggestion.

-      The second point ((“(ii)” in the paper, line 61) concerns the amount of TTV virions generated per day being 3.8 x 1010 per day. This number of produced virions per day has been calculated in one study on HCV infected patients only, and may require some extra study (e.g. in healthy persons). Suggestion is to delete this sentence as the amount of virus particles produced daily is not of relevance for the review.

We have deleted the sentence, as recommended.

-          The fourth point (“(iv)”, line 64) concerns compartmentalization. The studies referred to are quite old, and conducted at a time when the full width of anellomes were not known yet. It is still greatly unknown (as it is not carefully looked at) if compartmentalization truly occurs for human anelloviruses. Suggestion is to delete this sentence on compartmentalization as this is not of relevance for the review.

We have deleted the sentence, as recommended.